# Assessing the Efficacy of the Spectrum-Aided Vision Enhancer (SAVE) to Detect Acral Lentiginous Melanoma, Melanoma In Situ, Nodular Melanoma, and Superficial Spreading Melanoma

**DOI:** 10.3390/diagnostics14151672

**Published:** 2024-08-01

**Authors:** Teng-Li Lin, Chun-Te Lu, Riya Karmakar, Kalpana Nampalley, Arvind Mukundan, Yu-Ping Hsiao, Shang-Chin Hsieh, Hsiang-Chen Wang

**Affiliations:** 1Department of Dermatology, Dalin Tzu Chi General Hospital, No. 2, Min-Sheng Rd., Dalin Town, Chiayi 62247, Taiwan; tanglilin1121@hotmail.com; 2Institute of Medicine, School of Medicine, College of Medicine, National Yang Ming Chiao Tung University, No. 155, Sec. 2, Li-Nong Street, Beitou District, Taipei 112304, Taiwan; ctlu119@vghtc.gov.tw; 3Department of Surgery, Division of Plastic and Reconstructive Surgery, Taichung Veterans General Hospital, 1650 Taiwan Boulevard Sect. 4, Taichung 407219, Taiwan; 4Department of Mechanical Engineering, National Chung Cheng University, 168, University Rd., Min Hsiung, Chia Yi 62102, Taiwan; karmakarriya345@gmail.com (R.K.); nampellykalpana91@gmail.com (K.N.); d09420003@ccu.edu.tw (A.M.); 5Department of Dermatology, Chung Shan Medical University Hospital, No. 110, Sec. 1, Jianguo N. Rd., South Dist., Taichung City 40201, Taiwan; missyuping@gmail.com; 6Institute of Medicine, School of Medicine, Chung Shan Medical University, No. 110, Sec. 1, Jianguo N. Rd., South Dist., Taichung City 40201, Taiwan; 7Department of Surgery, Division of General Surgery, Kaohsiung Armed Forces General Hospital, 2, Zhongzheng 1st. Rd., Lingya District, Kaohsiung 80284, Taiwan; 8Department of Technology Development, Hitspectra Intelligent Technology Co., Ltd., Kaohsiung 80661, Taiwan

**Keywords:** skin cancer, acral lentiginous melanoma, melanoma in situ, modular melanoma, superficial spreading melanoma, hyperspectral imaging, band selection, spectrum-aided visual enhancer

## Abstract

Skin cancer is the predominant form of cancer worldwide, including 75% of all cancer cases. This study aims to evaluate the effectiveness of the spectrum-aided visual enhancer (SAVE) in detecting skin cancer. This paper presents the development of a novel algorithm for snapshot hyperspectral conversion, capable of converting RGB images into hyperspectral images (HSI). The integration of band selection with HSI has facilitated the identification of a set of narrow band images (NBI) from the RGB images. This study utilizes various iterations of the You Only Look Once (YOLO) machine learning (ML) framework to assess the precision, recall, and mean average precision in the detection of skin cancer. YOLO is commonly preferred in medical diagnostics due to its real-time processing speed and accuracy, which are essential for delivering effective and efficient patient care. The precision, recall, and mean average precision (mAP) of the SAVE images show a notable enhancement in comparison to the RGB images. This work has the potential to greatly enhance the efficiency of skin cancer detection, as well as improve early detection rates and diagnostic accuracy. Consequently, it may lead to a reduction in both morbidity and mortality rates.

## 1. Introduction

Skin cancer is the prevailing form of cancer in people and its occurrence is increasing significantly [1]. Skin cancers are classified based on the specific type of cell they originate from and their clinical characteristics [2]. Acral lentiginous melanoma is an uncommon kind of melanoma that primarily develops on the hands, feet, and nail beds [3]. While it is the most commonly identified subtype of the disease in many areas, especially in Latin America, Africa, and Asia, it is still relatively unexplored due to its distinct genetic causes [4]. Melanoma in situ refers to melanomas that are confined to the epidermis [5]. Despite being a rare type of skin cancer, its annual growth rate has climbed by 9.5%, making it one of the fastest-growing malignancies [6]. There have been no reported deaths directly caused by instances of melanoma in situ. However, people with this condition are at a significantly increased risk of getting another type of malignant melanoma (MM) in the future [7]. Cutaneous melanoma refers to a diverse group of malignant melanocytic growths that vary in terms of their epidemiology, morphology, growth patterns, genetics, and ability to spread to other parts of the body [8]. Nodular melanoma and superficial spreading melanoma are two prominent kinds of cutaneous melanoma. Nodular melanoma is a subtype of melanoma that is distinguished by its fast development rate. It is estimated that nodular melanoma invades the skin at a pace of approximately 0.5 mm per month [9].

Early diagnosis is the crucial determinant for the effective treatment of skin cancer [10]. A skin biopsy is a method of acquiring a sample from a suspected site of skin cancer, which is a difficult, time-consuming, and unpleasant procedure. Computer-aided diagnosis (CAD) offers a non-invasive, expedited, and real-time method for diagnosing skin cancer. Kousis et al. devised 11 advanced deep learning techniques to achieve a precise identification of skin cancer using a mobile application. Among these methods, the one utilizing DenseNet169 yielded the highest performance, with an accuracy of 92.25% and a recall rate of 93.59% [11]. Mehr et al. employed the inception-ResNet-v2 CNN deep learning architecture to identify skin cancer in lesion images, achieving an average accuracy of 89.3% [12]. Agrahari et al. developed a very efficient system for detecting skin cancer in many classes. They utilized a pre-trained MobileNet model on the HAM10000 ISIC dataset. The system demonstrated a level of performance that was comparable to that of a dermatological expert [13]. However, the majority of research primarily concentrates on RGB imaging, which utilizes only three distinct bands for picture processing. Nevertheless, the utilization of hyperspectral imaging (HSI) can greatly augment the quantity of data that can be acquired from each pixel, resulting in a greater level of precision.

The HSI technique is composed of numerous bands that have the ability to capture very detailed spectral responses that enable HSI to effectively identify even the most minor alterations in histopathological images [14]. An HSI image consists of pixels, each of which has a complete spectrum that makes it a highly effective tool for defining and studying biological substances [15]. In recent years, this new imaging modality has been utilized in medical applications [16,17,18,19]. Tsai et al. utilized HSI in conjunction with single shot multibox detector (SSD) architecture to detect esophageal cancer. Their study showed a significant improvement of over 5% in detection accuracy when HSI was employed instead of white-light imaging (WLI) [20]. HSI has not been fully employed in the field of skin cancer diagnosis within the bio-medical industry. The majority of studies primarily focus on detecting the most common types of skin cancer, such as basal cell carcinoma (BCC), squamous cell carcinoma (SCC), and seborrheic keratosis (SK), while paying less attention to the less common forms of skin cancer [21].

Narrow band imaging (NBI) is a specific sort of hyperspectral imaging (HSI) that captures images within a small range of wavelengths. This focused technique improves particular characteristics of an image, making it extremely efficient for medical diagnostics and other applications where precision and differentiation are essential [22]. This technology is based on the utilization of two short wavelength light beams, one at 415 nm (blue) and the other at 540 nm (green), to enhance optical images [23]. Longer wavelengths of light have a greater ability to enter tissues because of their scattering and absorption qualities. Specifically, 415 nm improves the visibility of superficial mucosal vascular forms, whereas 540 nm increases submucosal IPCLs [24]. This results in a noticeable difference in the picture, with the superficial vessels appearing in brown and the submucosal vessels appearing in cyan. This enhances the distinction between the blood vessels and the surrounding mucosa. [25]. Many studies have proven that NBI is better than white-light imaging (WLI) in terms of accuracy, sensitivity, and specificity [26,27,28,29,30].

Therefore, in this study, a novel method known as the spectrum-aided vision enhancer (SAVE) has been developed which has the ability to convert any WLI image into an HSI image and has been combined with band selection to select particular narrow bands. These SAVE images and WLI images of four different classes of skin cancers including acral lentiginous melanoma, melanoma in situ, nodular melanoma, and superficial spreading melanoma were trained using multiple YOLO architectures and the results were compared in terms of sensitivity, precision, F1-score, and accuracy.

## 2. Materials and Methods

### 2.1. Dataset

A total of 878 images was used in this study for the analysis of four different skin cancer classes including acral lentiginous melanoma with a total of 342 images, superficial spreading melanoma with a total of 253 images, nodular melanoma with a total of 100 images, and melanoma in situ with 183 images. The dataset was divided into training, testing, and validation in the ratio of 7:2:1, respectively, as shown in Table 1. These images were obtained from our partner hospital as well as from the International Skin Imaging Collaboration (ISIC), with equal contributions. Labelimg (version 1.8.6) software was used for the annotation and marking of the images. Image augmentation techniques such as clockwise and counter-clockwise rotation of 90 degrees and shear of 10 degrees in both vertical and horizontal orientations were also performed. Images were resized to 640 × 640 pixels to fit the YOLO architectures. The early disable function was also disabled to make the architectures run for 600 epochs fully, the batch size was set to 16, and value curves of the training and validation sets were determined after 600 epochs of training the loss function. 

ALM is represented by 231 close-up images, 12 dermoscopy images, and 102 clinical images. NM is represented by 61 close-up images, 16 dermoscopy images, and 26 clinical images. MIS is represented by 21 close-up images, 20 dermoscopy images, and 136 clinical images. SSM is represented by 35 close-up images, 34 dermoscopy images, and 184 clinical images. The distribution of images for each diagnosis group exhibits a notable disparity, which has the potential to induce bias in the model’s performance. The model is trained using a balanced strategy to mitigate the impact of any single modality on the outcomes, despite the variance in the quantity of images between modalities. This entails the utilization of normalization and augmentation strategies to alleviate the influence of features peculiar to each modality. The design of our model incorporates a wide range of images from multiple modalities, which improves its capacity to generalize across diverse input sources. The use of diverse imaging techniques in the model’s training data is advantageous rather than restrictive, as it equips the model to effectively handle real-world situations. The performance indicators, including as precision, recall, and mean average precision (mAP), are obtained through thorough validation and testing across all modalities. This guarantees that the model’s performance remains consistent and dependable, regardless of the type of image. Our methodology primarily emphasizes finding the fundamental illness patterns rather than surface-level picture attributes. The combination of the SAVE algorithm and the YOLO framework enhances the emphasis on pathological traits rather than artifacts peculiar to a certain modality (for a more detailed description of the dataset please see Appendix A).

### 2.2. YOLO Architectures

The choice of YOLO v5 for this experiment was made because of prior research findings that have shown its higher detection speed when compared with alternative architectures, such as Densenet or SSD. This supremacy is ascribed to its capacity to improve real-time performance. The YOLO v5 model includes a total of three modules: the neck, backbone, and head terminals. Convolutional Neural Networks (CNNs) often consist of architectural components such as focus, CONV-BN-Leaky ReLU (CBL), cross-stage partial (CSP), and spatial pyramid pooling (SPP) models [31]. Focus possesses the ability to combine multiple high-resolution images, divide the input image into smaller segments, reduce the required CUDA 11.8 memory and layers, enhance the rate of both forward and backward propagation, and extract image features. The provided image, which has a fixed size of 640 × 640 pixels, is split into four separate pictures, each measuring 320 × 320 pixels. After being partitioned, these images undergo CON-CAT, where they are combined with slices and layers of convolution kernels. This process produces an image with dimensions of 320 × 320 pixels. This partitioning strategy is utilized to optimize the efficacy of the training process. The SPP pooling layer is specifically designed to overcome the limitations of input size while preserving the integrity of the image. In this context, the term “neck” denotes a collection of layers that integrate characteristics from an image, resulting in the development of feature pyramid networks (FPN) and path aggregation networks (PAN) [32]. The system comprises the CBL, Upsample, CSP2_X, and several other models. YOLO v5 preserves the CSP1_X architecture from YOLO v4’s CSPDarknet-53 version and integrates the CSP2_X architecture to reduce the model’s dimensions and capture a more extensive range of image details. The GIoU Loss is a loss function utilized to quantify the difference between predicted and actual bounding boxes in the head. The loss function of YOLO v5 comprises three distinct types of losses: classification losses, confidence losses, and bounding box regression losses [33]. The loss function quantifies the discrepancy between the observed and predicted results of the model. The YOLO v5 model’s loss function is defined as follows: (1)$$L_{GIOU}=\sum_{i=0}^{S^{2}}\sum_{j=0}^{B}I_{ij}^{obj}\left[1-IOU+\frac{A^{c}-U}{A^{c}}\right]$$
(2)$$-\lambda_{noobj}\sum_{i=0}^{S^{2}}\sum_{j=0}^{B}I_{ij}^{noobj}\left[{\hat{c}}_{i}^{j}log\left(C_{i}^{j}\right)+\left(1-C_{i}^{j}\right)log\left(1-C_{i}^{j}\right)\right]$$
(3)$$L_{class}=-\sum_{i=0}^{S^{2}}I_{ij}^{noobj}$$
(4)$$\sum_{c\in \mathrm{classes}}\left[{\hat{P}}_{i}^{j}\left(c\right)log\left(P_{i}^{j}\left(c\right)\right)+\left(1-{\hat{P}}_{i}^{j}\left(c\right)\right)log\left(1-P_{i}^{j}\left(c\right)\right)\right]$$

In January 2023, Ultralytics (Los Angeles, CA, USA) confirmed the newest member of the YOLO family, YOLO v8 [34]. The YOLO v8 concept does not rely on anchors. This suggests that instead of making a prediction about the distance between an item and a known anchor box, it directly calculates the center of the object. Anchor-free detection reduces the need for predicting bounding boxes, hence accelerating Non-Maximum Suppression (NMS), a complex post-processing step that filters potential detections after inference. The fundamental unit was modified, with C2f substituting C3, and the original 6 × 6 convolution of the stem being substituted with a 3 × 3. YOLO v8 utilizes the PAN-FPN approach in the neck section to enhance the fusion and exploitation of feature layer information at various scales [35,36]. YOLO v8 utilizes two methods for increasing resolution and many C2f modules in combination with the ultimate decoupled head structure to create the neck module. It also incorporates the concept of CSP from YOLO v5 [37]. YOLO v8 includes the concept of separating the head in YOLOx and applies it to the latter half of the neck. The integration of confidence and regression boxes results in a heightened level of accuracy.

Wang et al. released YOLO v9, the most recent version of the YOLO family of real-time object identification models, in February 2024 [38]. This marks a notable progression in the field of object detection, with the goal of surpassing methods based on convolution and transformers. This model incorporates programmable gradient information (PGI) and the Generalized Efficient Layer Aggregation Network (GELAN) to improve accuracy [39]. This type is specifically intended for object detection jobs and is renowned for its exceptional efficiency in real-time detection, offering enhanced accuracy and speed [40]. YOLO v9 has been evaluated against its predecessor, YOLO v8, in multiple tests, showcasing its higher accuracy while keeping the same training efficiency. Although YOLO v9 has a longer training period, it provides a favorable balance between accuracy and efficiency, making it a potential option for applications that need accurate object detection. The performance of the model in reliably detecting objects has been thoroughly analyzed in terms of precision, recall, F1-score, and loss functions.

### 2.3. Spectrum-Aided Vision Enhancer

This study devised a VIS-HSI transformation mechanism called SAVE (developed by Hitspectra Intelligent Technology Co., Ltd., Kaohsiung City, Taiwan), which can convert an RGB image taken by an electronic camera into an HSI image, as illustrated in Figure 1. Firstly, it is essential to determine the relationship among an RGB picture and the spectrometer’s readings for different colors. The specified colors for calibration are identified as the X-Rite Classic. The tool consists of 24 squares displaying a diverse range of color samples commonly found in natural surroundings. The samples include the colors red, green, blue, cyan, magenta, and yellow, as well as six different hues of gray. X-Rite has gained popularity as a preferred option for color calibration in recent years. The main objective of the camera was to capture photographs that accurately represented the colors of the Macbeth Color Checker, which acted as the reference point. The patch image, which contained 24 colors, was transformed to the CIE 1931 XYZ color space. The digital camera recorded an image that was subsequently stored in JPEG, using the standard RGB (sRGB) color scheme. The first stage entailed converting the R, G, and B values (ranging from 0 to 255) within the sRGB color space into a more limited range that spans from 0 to 1. Later on, the Gamma function was employed to transform the normalized sRGB characteristics into calibrated RGB values. The RGB values were converted to the CIE 1931 color space using a transformation matrix. This matrix depicts the quantitative relationship among the wavelengths in the visible spectrum (VIS) and the perceived color in the natural world. However, the images captured with a digital camera might be affected by various causes such as nonlinear response, dark current, improper color separation, or color shift. Thus, Equation (5), which represents a matrix with varying values, was utilized. Equation (6) was utilized to compute the calibrated values of X, Y, and Z, known as XYZ_correct_, as part of the error correction procedure. (the equations for modifying the 24-color patch picture and 24-color reflectance spectrum information to transform them into the XYZ color space are explained in Appendix A).
(5)$$\left[C\right]=\left[{XYZ}_{Spectrum}\right]\times pinv(\left[V\right])$$
(6)$$\left[{XYZ}_{Corrcnt}\right]=\left[C\right]\times [V]$$

The brightness ratio was determined by utilizing the *Y* value of the XYZ color gamut space, as this parameter exhibits a direct correlation with brightness. The data obtained from the reflectance spectrum were converted into the XYZ value (*XYZ_Spectrum_*), which was then standardized to suit the XYZ color gamut space. The correction coefficient matrix *C* was derived through the application of multiple regression analysis, as described in Equation (3). The transformation matrix (*M*) was derived using the reflectance spectrum data (*R_spectrum_*). Principle component analysis (PCA) was conducted on the *R_spectrum_* dataset in order to obtain 6 significant principal components (PCs) and eigenvectors. These components were able to account for 99.64% of the information. The average root mean square error (RMSE) between the *XYZ_correct_* and *XYZ_Spectrum_* models for the 24 target colors was 0.19, indicating a minor difference. Next, a multiple regression analysis (MRA) which is a method of statistical analysis that can be employed to examine the correlation between one dependent variable and multiple independent variables was conducted on *M* and correlated with the six significant PCs. These PCs were utilized in the multivariate regression analysis of *XYZ_correct_*. The variable *V_color_* was deliberately selected due to its comprehensive coverage of all permutations of the X, Y, and Z values. The calibration of the camera is a crucial component of the SAVE algorithm. Following this, the color exhibited a striking resemblance to the color acquired by the spectrum analyzer, rendering the distinction challenging to perceive. Prior to camera calibration, the mean chromatic aberration of all 24 color blocks was 10.76. After calibrating, the mean chromatic aberration decreased to a minimum of 0.63. The analog spectrum (*S_Spectrum_*) was computed from *XYZ_correct_* using Equation (4). Next, the *S_Spectrum_* was assessed in comparison to the *R_spectrum_*. The RMSE for the 24 color blocks was found to be 0.056, while the mean color variation between the analog spectrum and reflectance spectrum produced from the spectrometer was determined to be 0.75. This observation suggests that the colors derived from the reflectance spectrum closely resembled the colors associated with the observed values. Hence, employing the aforementioned methodology, it is possible to convert an RGB image into an HSI image.

The spectrometer employed for this study was the Ocean Optics QE65000, which produces a reflectance spectrum of a 24-color patch. The brightness ratio was calculated by using the value for Y of the XYZ color gamut space, as this variable has a direct association with brightness. The reflectance spectrum data were transferred into the XYZ value (*XYZ_Spectrum_*), and subsequently adjusted to conform to the XYZ color gamut space. The correction coefficient matrix *C* was calculated by applying MRA, as outlined in Equation (7). The transformation matrix (*M*) was obtained by utilizing the reflectance spectrum data (*R_spectrum_*). PCA was conducted on the *R_spectrum_* dataset to identify the 6 most relevant principal components (PCs) and their corresponding eigenvectors. These components were capable of explaining 99.64% of the data being analyzed. The RMSE comparing the *XYZ_correct_* and *XYZ_Spectrum_* estimates for the 24 target colors had an average value of 0.19, suggesting a negligible disparity. Subsequently, an MRA was conducted on variable *M* and its correlation was examined with the six significant principal components. The PCs were employed in the multivariate regression analysis of *XYZ_correct_*. The variable *V_color_* was intentionally chosen since it encompasses every possible combination of the X, Y, and Z values. The calibration of the camera is an essential element of the SAVE algorithm. After calibrating the camera, the color closely matched the color obtained by the spectrum analyzer, making it difficult to distinguish between them. Before calibrating the camera, the mean chromatic aberration of all 24 color blocks was 10.76. After calibrating the camera, the chromatic aberration was reduced to a minimum value of 0.63. Equation (8) was used to calculate the analog spectrum (*S_Spectrum_*) from *XYZ_correct_*. Subsequently, the *S_Spectrum_* was evaluated in relation to the *R_spectrum_*. The RMSE for the 24 color blocks was calculated to be 0.056. Additionally, the average color discrepancy within the *S_Spectrum_* and *R_spectrum_* generated by the spectrometer was measured to be 0.75. This finding indicates that the colors obtained from the reflectance spectrum closely matched the colors linked to the observed values. Therefore, by utilizing the previously indicated approach, it is feasible to transform an RGB image into an HSI image.
(7)$$\left[M\right]=\left[Score\right]\times pinv(\left[V_{Color}\right])$$
(8)$${\left[S_{Spectrum}\right]}_{380\sim 780\;nm}=\left[EV\right]\left[M\right][V_{Color}]$$

### 2.4. Band Selection

Although the SAVE algorithm has the ability to convert the RGB images into HSI images, suitable narrow-bands must be selected from the spectrum of visible bands to enhance the diagnostic accuracy. For this, in this study, the narrow bands used in the Olympus endoscope were simulated, calculated, and replicated. The simulated NBI image from the SAVE conversion algorithm must be similar to the real NBI image captured by Olympus endoscope. Also, for this calibration, the standard 24-color checker was used. The NBI image simulated from the HSI conversion algorithm was compared with the real NBI image captured by the Olympus (Tokyo, Japan) endoscope. The CIEDE 2000 color difference between each of the 24 color blocks was measured and minimized. After the correction, the average color difference of the 24 color blocks was found to be only 2.79, which is negligible. Although the peak absorption wavelengths of hemoglobin are 415 nm and 540 nm, the real NBI image taken by the Olympus endoscope includes not only green and blue colors, but also various shades of brown, which correspond to a wavelength of 650 nm. Figure 2 displays the contrast of the wavelength between SAVE, Olympus WLI, and Olympus NBI. Thus, it can be inferred that there is a nuanced image post-processing technique that enhances the realism of NBI videos. Therefore, in this study, in addition to the wavelengths of 415 nm and 540 nm, three other regions at 600 nm, 700 nm, and 780 nm also exhibit a light spectrum. The entirety of the process is shown in Figure 3.

### 2.5. Evaluation Indices

The diagnostic performance of the YOLO architectures was assessed in this study using multiple factors, including precision, recall, specificity, accuracy, and F1 score. Accuracy, in the context of machine learning, is a quantitative metric that evaluates the proportion of accurate predictions made by the model compared to the total number of predictions. The calculation involves evaluating the proportion of accurate predictions to the overall number of predictions [41]. A higher degree of precision indicates a model that exhibits superior performance in terms of overall accuracy. Precision is a numerical measure that determines the proportion of accurate positive predictions made by a model out of the overall amount of positive predictions. The calculation entails the division of the total amount of true positives by the sum of precise positives and false positives [42]. Precision is a metric that quantifies the accuracy of positive predictions by measuring the percentage of predicted positive instances that are truly correct. Recall, also known as sensitivity, is a quantitative metric that evaluates the proportion of accurate positive predictions made by a model compared to the overall amount of actual positive occurrences. The calculation requires dividing the total amount of true positives by the sum of true positives and false negatives. The concept of recall emphasizes the model’s ability to correctly identify positive instances from the overall amount of actual positives [43]. The F1-score is a quantitative metric that quantifies the harmonic average of precision and recall. The purpose of this is to achieve a fair assessment between these two metrics [44]. The calculation is obtained by multiplying the values of precision and recall and then multiplying the resulting product by 2. The product is then divided by the total number of precision and recall values. The F1-score is a useful metric for evaluating the effectiveness of the architecture in classification tasks as it considers both false positives and false negatives. Mean Average Precision (mAP) is a crucial metric employed in object detection tasks to evaluate the precision of models in detecting and classifying objects in images [45]. The calculation involves determining the average precision (AP) for all recall values ranging from 0 to 1. mAP, which stands for mean Average Precision, is a commonly used metric in the field of computer vision.
(9)$$Accuracy=\frac{TP+TN}{TP+TN+FP+FN}\times 100$$
(10)$$Precision=\frac{TP}{TP+FP}\times 100$$
(11)$$Recall=\frac{TP}{TP+FN}\times 100$$
(12)$$F1-score=2\times \frac{P\times R}{P+R}\times 100$$

*TP* = True positive; *TN* = True negative; *FN* = False negative; *FP* = False positive; *P* = Precision; and *R* = Recall.

## 3. Results

### 3.1. SAVE Performance Evaluation

The data generated through simulation were utilized for the initial evaluation of the system’s performance. The characterization involved measuring their spectrum emissions, while also taking into account the sensitivities provided by the camera manufacturers. The spectral curves of the 24-color Macbeth Color Checker chart were utilized for both the training and validation sets of the samples. Table 2 displays the RMSEs and standard deviation (SD) values of the XYZ color space coordinates before and after calibration for a total of 24 colors. RMSE measures the average difference between anticipated and actual values. A lower RMSE number indicates a higher level of accuracy in the calibration process. The XYZ values denote color coordinates that are crucial for precise image representation. SD quantifies the dispersion of these numbers, where a smaller SD indicates more consistency and reliability of the data. Table 2 illustrates the efficacy of the calibration method in diminishing mistakes and enhancing the accuracy of color depiction in photographs. Calibrating the SAVE is essential to ensure the dependability of its ability to convert RGB photos into HSI-NBI for precise skin cancer detection.

The calibration of the camera is a crucial component of the SAVE algorithm. Figure 4 displays the color disparity outcomes prior to and following calibration. Following the calibration of the camera, the color exhibited a striking resemblance to the color acquired by the spectrum analyzer, rendering the distinction challenging to perceive. Prior to camera calibration, the mean chromatic aberration of all 24 color blocks was 10.76. After calibrating the camera, the average chromatic aberration decreased to a minimum of 0.63.

In Figure 5, the reflectance values of the six primary colors inside the 24-color block are illustrated. These colors are blue (13), red (15), green (14), yellow (16), magenta (17), and cyan (18). Based on the analysis of the 24 color blocks, it was noted that the red block exhibited the most significant disparity between the simulated and actual reflectance values, particularly across the longer wavelength range of 600 to 780 nm. One of the limitations of the study is considered to be this element. All of the remaining 23 color blocks exhibited RMSEs below 0.1, with the color black demonstrating the lowest RMSE of 0.015. The RMSE was merely 0.056, suggesting that the majority of the color could be replicated with precision.

RMSE values can be graphically and mathematically represented by calculating the disparity between simulated and measured colors. The representation of a color can be denoted as LAB, where L represents lightness, channel A, and channel B, respectively. The numerical definition of any color can be achieved by manipulating the values of L, A, and B. The L, A, and B values of the simulated and computed colors are depicted in Figure 6. The average color disparity was about 0.75, suggesting that the replicated color was visually precise.

### 3.2. Model Performance Evaluation

In this study examining the efficacy of the SAVE algorithm using different YOLO machine learning models and the Roboflow 3.0 model, a comprehensive analysis was performed focusing on metrics like precision, recall, mAP, and F1-score are shown in Table 3 and Table 4. The YOLO v8 model emerged as the most effective, outperforming other models such as YOLO v9, YOLO v5, YOLO-NAS, and Roboflow 3.0 in various aspects of object detection. This superior performance is highlighted by its higher mAP score, which reflects better overall prediction accuracy across multiple threshold settings. The superior precision of the YOLO v8-SAVE model, achieving above 90%, contrasts with a lower recall of 71%, indicating that while the model is highly accurate in detecting true positives, it falls short in identifying all actual positives. This is further underscored by comparing it to the YOLO v8-RGB model, which balanced precision and recall more evenly at 84.3% and 75.5%, respectively. This discrepancy suggests that the YOLO v8 model, when configured for the SAVE algorithm, prioritizes accuracy over coverage, potentially overlooking some true positive cases in favor of maintaining high confidence in the cases it does detect. To address this imbalance, particularly the lower recall, adjustments such as modifying the decision threshold, enhancing feature selection, or enriching the training data with more diverse examples could be beneficial. These changes might help the model better identify subtle features indicative of positive cases, thereby increasing the recall without sacrificing precision significantly. YOLO v5, while second in overall performance, demonstrated an interesting pattern where the recall was higher relative to its precision. This suggests that YOLO v5 may be better suited in scenarios where missing a true positive is more detrimental than misclassifying a negative as a positive. The high mAP of 81.9% for the YOLO v5-SAVE model indicates robustness in its predictive capability across various thresholds, although it trails behind YOLO v8 in precision and overall performance. The YOLO v9 model had a similar result in comparison with the YOLO v5 model in terms of precision and mAP. However, both YOLO v5 and YOLO v9 had a better recall and F1-score. The YOLO-NAS model showed comparatively poor results in precision, recall, and mAP, highlighting possible limitations in its architecture or training data that could be impeding its performance. This might require a reevaluation of the model’s structural parameters or an enhancement of its learning algorithm to better capture the necessary features for accurate diagnosis.

In terms of precision, acral lentiginous melanoma and nodular melanoma, in general, showed higher precision in the SAVE model in both YOLO v5 and YOLO v8 models while melanoma in situ and superficial spreading melanoma had inconsistent results, with a decrease of almost 23% and 7% in the YOLO v5-SAVE model and an 18% and 1% increase in YOLO v8 respectively. In terms of recall across the two ML architectures, the RGB model had either better or similar results compared to SAVE except in two occasions where SAVE outperformed the WLI model. YOLO v9 had a different pattern of results with acral lentiginous melanoma being the lowest in terms of precision with 65.7% in the RGB model, which was improved 29% in the SAVE model, and melanoma in situ had a higher precision of 94.9% in the RGB model which decreased to 79.8% in the SAVE model. However, similar to the other models, the recall values when compared with the precision were less, although the average mAP50 in the YOLO v9 model was found to be higher. This finding showcased the model’s exceptional level of accuracy in learning lesion features. The accuracy rates of both the RGB model and the SAVE model appear to be similar due to the relatively smaller number of images used. Nevertheless, this study demonstrates that the SAVE-based conversion algorithms, which possess the ability to transform RGB images into HSI images, have the potential to accurately classify and detect skin cancers. While the majority of the findings are comparable, this study has notably enhanced the accuracy rate as shown in Figure 7.

## 4. Discussion

Given these varied performances, the choice of model can be strategic depending on the specific requirements of the medical diagnostic task. For instance, in settings where missing a diagnosis could lead to severe outcomes, a model with higher recall like YOLO v5 might be preferable. Conversely, in situations where precision is paramount to avoid unnecessary treatments, YOLO v8 would be more appropriate. Thus, selecting the optimal model involves a trade-off between these metrics, tailored to the needs of the medical application. This analysis not only underscores the strengths and weaknesses of each model but also offers a pathway to refine these tools for enhanced diagnostic accuracy in medical imaging. One key limitation of this study is the potential lack of generalizability due to the specific dataset used, which might not adequately represent the diverse range of skin types and cancer stages. Additionally, the study primarily focused on quantitative metrics such as precision and recall, potentially overlooking qualitative aspects like model interpretability and clinical usability. The models were evaluated in controlled experimental settings, which may not fully capture their performance in real-world clinical environments. Lastly, the computational demands of these advanced models might limit their practical application in resource-constrained settings, necessitating further optimization for broader deployment. To overcome the limitations identified in this study, future research will prioritize diversifying the dataset to include a wider range of skin types, conditions, and stages of cancer to enhance the models’ generalizability. Incorporating qualitative assessments such as user feedback from clinicians can help refine the models for better usability and interpretability in real-world settings. Additionally, developing lightweight versions of these models or leveraging newer, more efficient machine learning frameworks could reduce computational demands, making the technology more accessible in resource-limited environments. Collaboration with clinical partners for pilot testing can also provide critical insights into practical challenges and user requirements.

The future scope of this study in evaluating the SAVE algorithm through various YOLO models and the Roboflow 3.0 framework in skin cancer detection is expansive and promising. Given the preliminary findings, future research can explore several avenues to enhance the accuracy and applicability of these models. Firstly, refining the SAVE algorithm with advanced machine learning techniques such as deep learning ensemble methods could further improve the precision and recall metrics, potentially leading to better diagnostic outcomes. Exploring the integration of additional imaging modalities alongside hyperspectral imaging could also provide richer data for model training and validation, increasing the robustness of the diagnostic system. Moreover, the application of the models in real-world clinical settings as part of routine skin cancer screenings could provide valuable feedback on their operational effectiveness and user-friendliness. This practical deployment can help identify unforeseen challenges and user requirements, leading to iterative model improvements. Additionally, expanding the dataset to include a broader spectrum of skin types, conditions, and stages of cancer will be crucial for ensuring the models’ generalizability and sensitivity across diverse patient populations. Another significant area for future research is the development of real-time diagnostic tools using these models. Leveraging the real-time processing capabilities of YOLO architectures can facilitate the development of mobile or handheld devices for immediate skin cancer screening and diagnosis, making effective use of the models’ ability to provide instant results. These efforts combined could vastly improve the early detection and treatment of skin cancer, ultimately saving lives and reducing healthcare costs.

## 5. Conclusions

This study has effectively demonstrated the superior performance of the YOLO v8 model integrated with the SAVE algorithm for detecting skin cancer, surpassing other models like YOLO v5, YOLO-NAS, YOLO v9, and Roboflow 3.0 in terms of precision, recall, mAP, and F1-score. Despite its impressive precision, YOLO v8 showed a comparatively lower recall, suggesting a need for adjustments in model threshold and training strategies to better balance detection accuracy with the ability to identify all relevant cases. YOLO v5 also showed promising results, particularly in recall, indicating its potential utility in scenarios where higher sensitivity is required. The study highlights the importance of tailored model selection based on specific diagnostic needs and sets the stage for the further refinement and real-world application of these advanced machine learning tools in the medical field. Future research should focus on enhancing model generalizability, exploring additional imaging modalities, and expanding the deployment of these technologies in clinical settings.

## Figures and Tables

**Figure 1 diagnostics-14-01672-f001:**
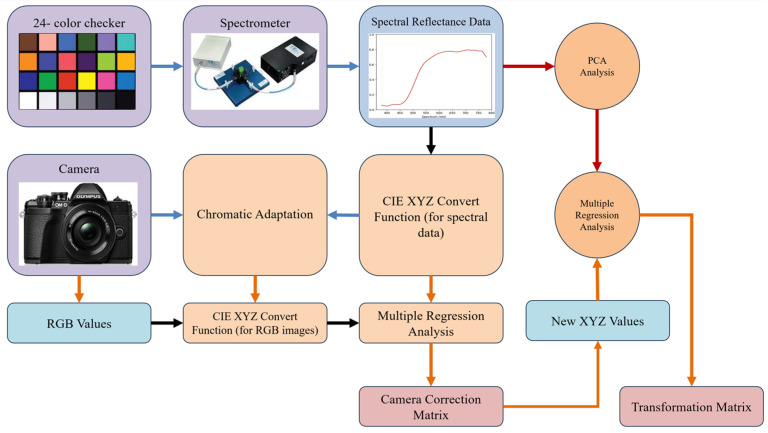
SAVE flowchart.

**Figure 2 diagnostics-14-01672-f002:**
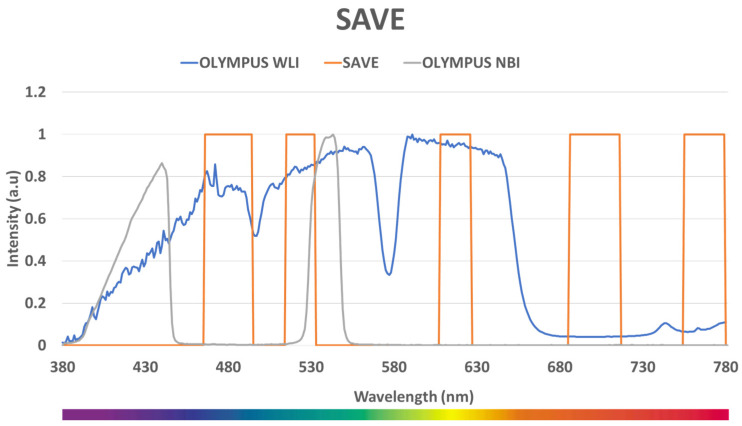
Comparison of wavelengths of Olympus WLI, Olympus NBI, and SAVE.

**Figure 3 diagnostics-14-01672-f003:**
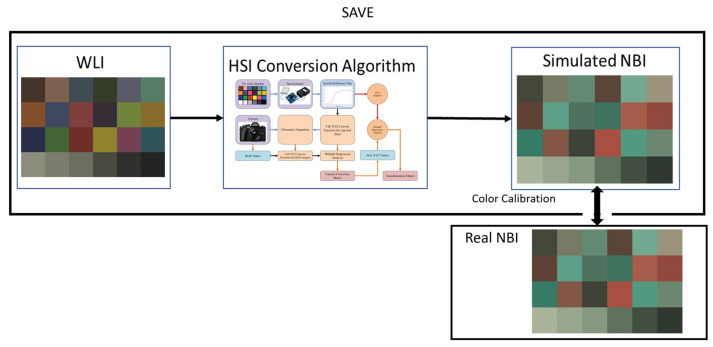
Flowchart of the procedure for band selection.

**Figure 4 diagnostics-14-01672-f004:**
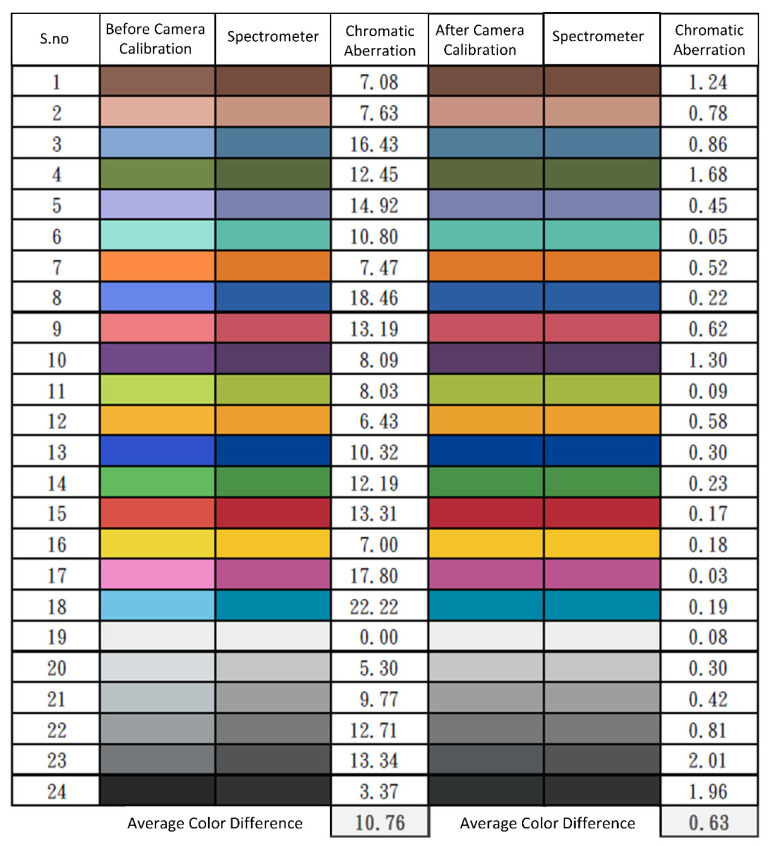
Color difference before and after camera calibration.

**Figure 5 diagnostics-14-01672-f005:**
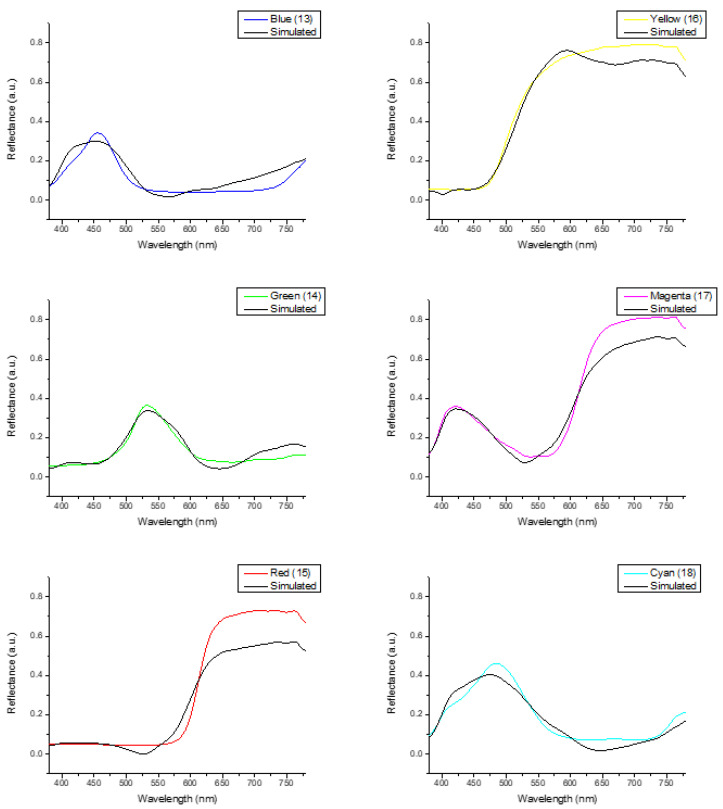
RMSEs between analog and measured spectra of each color block.

**Figure 6 diagnostics-14-01672-f006:**
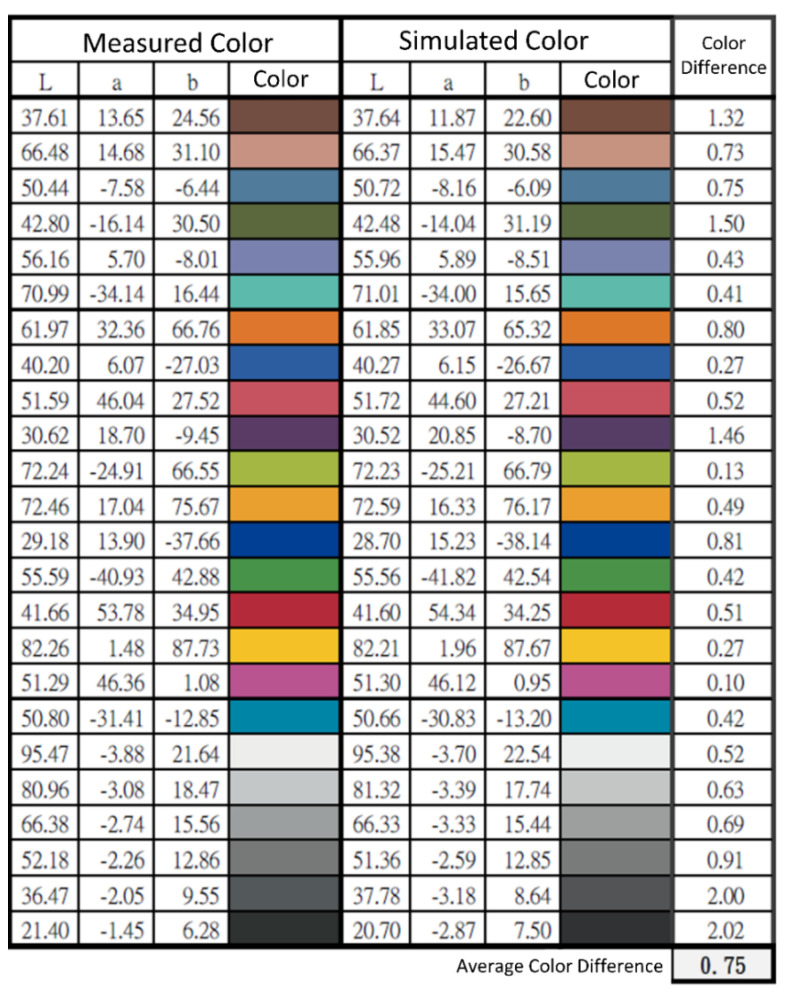
LAB values of the simulated and observed colors.

**Figure 7 diagnostics-14-01672-f007:**
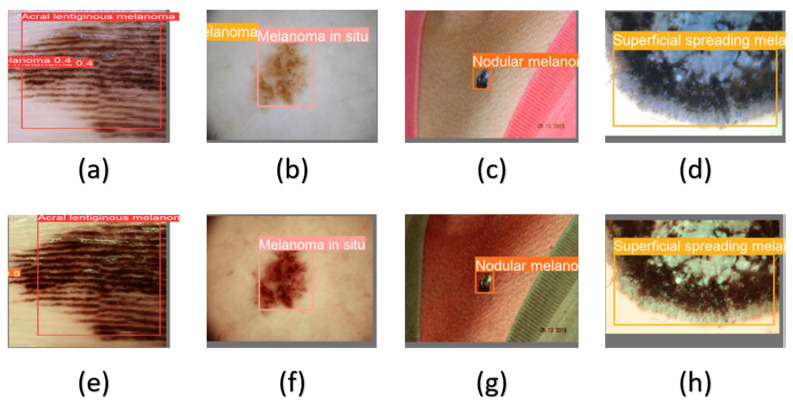
Different classes of skin cancer used in this study. (**a**) and (**e**) represent acral lentiginous melanoma in RGB and SAVE, respectively; (**b**) and (**f**) represent melanoma in situ in RGB and SAVE, respectively; (**c**) and (**g**) represent nodular melanoma in RGB and SAVE, respectively; and (**d**) and (**h**) represent superficial spreading melanoma in RGB and SAVE, respectively.

**Table 1 diagnostics-14-01672-t001:** Dataset used in this study.

Classification	Training	Validation	Testing
Acral Lentignious Melanoma	239	69	34
Melanoma in Situ	128	37	18
Nodular Melanoma	70	20	10
Superficial Spreading Melanoma	178	50	25
Total	615	176	87

**Table 2 diagnostics-14-01672-t002:** RMSEs of the XYZ values before and after calibration.

S. No	Before Calibration			After Calibration			RMSE	SD
	X	Y	Z	X	Y	Z		
1	10.96	9.92	4.63	11.14	9.87	4.26	0.24	0.30
2	38.74	35.80	18.65	38.57	35.94	18.66	0.13	0.08
3	16.62	19.07	24.13	16.48	18.79	24.11	0.18	0.17
4	10.33	12.86	4.62	10.16	13.03	4.85	0.19	0.19
5	24.05	23.87	31.55	24.16	24.07	31.60	0.13	0.08
6	30.12	42.15	32.40	30.10	42.17	32.42	0.02	0.002
7	38.10	30.24	4.28	38.04	30.37	4.22	0.09	0.04
8	11.70	11.47	25.90	11.64	11.37	25.91	0.07	0.02
9	29.01	19.91	9.62	29.20	19.78	9.60	0.13	0.08
10	8.26	6.49	9.63	8.06	6.49	9.86	0.18	0.17
11	34.15	44.06	8.44	34.15	44.02	8.53	0.06	0.01
12	47.99	44.55	6.05	48.05	44.34	6.17	0.15	0.11
13	6.82	5.79	21.07	6.90	5.91	21.00	0.09	0.04
14	14.55	23.55	7.22	14.58	23.51	7.12	0.07	0.02
15	21.08	12.25	3.57	21.01	12.28	3.65	0.06	0.01
16	58.40	60.69	7.54	58.38	60.79	7.42	0.09	0.04
17	28.98	19.54	20.67	28.94	19.52	20.66	0.02	0.002
18	12.81	19.01	28.54	12.84	19.10	28.56	0.05	0.01
19	82.12	88.54	67.20	82.31	88.73	67.51	0.24	0.30
20	54.74	58.92	45.52	54.28	58.40	44.75	0.60	1.89
21	33.08	35.73	27.24	33.26	35.82	27.54	0.21	0.23
22	18.18	19.62	14.94	18.86	20.31	15.62	0.68	2.43
23	9.13	10.01	8.13	8.56	9.26	7.21	0.76	3.04
24	2.87	3.19	2.39	3.10	3.35	2.68	0.23	0.27
Average							0.19	0.39

**Table 3 diagnostics-14-01672-t003:** Evaluation of different ML models with respect to the two imaging modalities.

Model	Image Modality	Precision	Recall	mAp	F1-Score
YOLO v5	RGB	0.804	0.716	0.797	0.751
	SAVE	0.799	0.829	0.819	0.810
YOLO v8	RGB	0.843	0.755	0.807	0.795
	SAVE	0.904	0.710	0.801	0.794
YOLO v9	RGB	0.806	0.605	0.737	0.65
	SAVE	0.783	0.666	0.775	0.71
YOLO-NAS	RGB	0.733	0.541	0.659	0.623
	SAVE	0.731	0.665	0.690	0.69
Roboflow 3.0	RGB	0.719	0.643	0.660	0.675
	SAVE	0.781	0.613	0.680	0.68

**Table 4 diagnostics-14-01672-t004:** Precision, recall, mAP of the four different classes of skin cancer studies in YOLO v5 and YOLO v8.

Architecture	Model	Skin Cancer Types	Precision	Recall	mAP50	mAP 50–95
YOLO v-5	RGB	Acral Lentiginous Melanoma	0.865	0.875	0.966	0.6
		Melanoma in Situ	0.78	0.545	0.597	0.323
		Nodular Melanoma	0.732	0.9	0.932	0.679
		Superficial Spreading Melanoma	0.784	0.542	0.612	0.33
	SAVE	Acral Lentiginous Melanoma	0.919	0.812	0.903	0.551
		Melanoma in Situ	0.551	0.455	0.481	0.223
		Nodular Melanoma	0.84	0.8	0.842	0.6
		Superficial Spreading Melanoma	0.669	0.625	0.598	0.299
YOLO v-8	RGB	Acral Lentiginous Melanoma	0.938	0.95	0.965	0.562
		Melanoma in Situ	0.685	0.545	0.608	0.294
		Nodular Melanoma	0.867	0.9	0.932	0.62
		Superficial Spreading Melanoma	0.882	0.623	0.725	0.419
	SAVE	Acral Lentiginous Melanoma	0.946	0.812	0.957	0.55
		Melanoma in Situ	0.859	0.555	0.7	0.286
		Nodular Melanoma	0.851	0.9	0.891	0.631
		Superficial Spreading Melanoma	0.875	0.582	0.639	0.338
YOLO v-9	RGB	Acral Lentiginous Melanoma	0.657	0.846	0.814	0.508
		Melanoma in Situ	0.949	0.667	0.826	0.46
		Nodular Melanoma	0.882	0.231	0.533	0.33
		Superficial Spreading Melanoma	0.737	0.676	0.776	0.472
	SAVE	Acral Lentiginous Melanoma	0.849	1	0.995	0.4
		Melanoma in Situ	0.798	0.607	0.749	0.45
		Nodular Melanoma	0.755	0.571	0.765	0.416
		Superficial Spreading Melanoma	0.73	0.484	0.579	0.31

## Data Availability

The data presented in this study are available from the corresponding author (H.-C.W.) upon reasonable request.

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
