# Peer review of "Assessing the Efficacy of the Spectrum-Aided Vision Enhancer (SAVE) to Detect Acral Lentiginous Melanoma, Melanoma In Situ, Nodular Melanoma, and Superficial Spreading Melanoma"

_diagnostics, 2024, doi:10.3390/diagnostics14151672_

Round 1
Reviewer 1 Report
Comments and Suggestions for Authors
In their manuscript, the authors present a novel image colorspace processing technique, leading to an improved skin cancer classification using artificial neural networks.
The paper is well written and highlights a new, relevant technology for potentially improving skin cancer diagnostic accuracy using AI.
However, this study (or at least the report) suffers from a serious methodological flaw: In figure 7, multiple image modalities (clinical, close-up and dermoscopy) are shown, with most diagnosis groups having a distinct modality. In the methods part, only "images" are mentioned. I could not find any mention or discussion of the different imaging modalities and their distribution within the dataset, which in my opinion is needed. Depending of the imbalance in the distribution, this could lead to a smaller or bigger bias in the model's performance. If, for example, all pictures of acral lentiginous melanoma are close-up's vs. all pictures of melanoma in situ being dermscopic, the model could learn the difference in picture structure due to the different modalities and not the underlying disease pattern. So at the very least, this distribution should be included and discussed.
That being, said, exploring colorspace preprocessing for improving the accuracy of skin cancer diagnostics surely is an innovative and interesting avenue.
Reviewer 2 Report
Comments and Suggestions for Authors
The study aims to evaluate the effectiveness of spectrum-aided visual enhancers using 878 images. The authors also included mathematical methods in the study. Table 2. RMSEs of the XYZ values before and after calibration are given for 24 variables. The table has two different things like 24 variables and X, Y, Z and relevant SD values are given. Now how the data should be considered to the read, is not clear. The reliability of the data in Table - 4 should be verified. Overall, the manuscript is impressive but the data given in the manuscript should be checked for its reliability.
